# Exercise-Mediated Protection against Air Pollution-Induced Immune Damage: Mechanisms, Challenges, and Future Directions

**DOI:** 10.3390/biology13040247

**Published:** 2024-04-08

**Authors:** Xingsheng Jin, Yang Chen, Bingxiang Xu, Haili Tian

**Affiliations:** School of Exercise and Health, Shanghai University of Sport, Shanghai 200438, China; jin_9919@163.com (X.J.); chenyang99202202@163.com (Y.C.)

**Keywords:** air pollution, exercise, immune health, protective mechanisms

## Abstract

**Simple Summary:**

This review explored the combined impacts of air pollution and exercise on the immune system. We proposed the possible mechanisms and evidence of their effects on immune health in the body through population, animal, and cell experiments. Research on the mechanisms by which exercise counteracts the damage to immune health caused by air pollution is still in its initial stages. Challenges and directions for determining the mechanism of exercise against air pollution-induced damage are proposed.

**Abstract:**

Air pollution, a serious risk factor for human health, can lead to immune damage and various diseases. Long-term exposure to air pollutants can trigger oxidative stress and inflammatory responses (the main sources of immune impairment) in the body. Exercise has been shown to modulate anti-inflammatory and antioxidant statuses, enhance immune cell activity, as well as protect against immune damage caused by air pollution. However, the underlying mechanisms involved in the protective effects of exercise on pollutant-induced damage and the safe threshold for exercise in polluted environments remain elusive. In contrast to the extensive research on the pathogenesis of air pollution and the preventive role of exercise in enhancing fitness, investigations into exercise resistance to injury caused by air pollution are still in their infancy. In this review, we analyze evidence from humans, animals, and cell experiments on the combined effects of exercise and air pollution on immune health outcomes, with an emphasis on oxidative stress, inflammatory responses, and immune cells. We also propose possible mechanisms and directions for future research on exercise resistance to pollutant-induced damage in the body. Furthermore, we suggest strengthening epidemiological studies at different population levels and investigations on immune cells to guide how to determine the safety thresholds for exercise in polluted environments.

## 1. Introduction

The World Health Organization (WHO) updated its Air Quality Guidelines on 21 September 2021, setting new levels for six major pollutants (CO, lead, NO_2_, O_3,_ PM, and SO_2_) [1]. These guidelines aim to protect human health from air pollution, which causes about 7 million premature deaths worldwide, every year. However, according to the WHO data, nearly everyone on Earth (99%) breathes air that exceeds the limits and contains high levels of pollutants. Low- and middle-income countries are particularly affected. Air pollutants can impair the body’s natural immune system by increasing oxidative stress and inflammation. A well-functioning immune system is essential for maintaining health and preventing diseases. The immune system must balance adequate and excessive responses to various challenges. Exposure to air pollutants can disrupt this balance by inhibiting the activity of relevant immune cells and causing abnormal responses. This can lead to immune damage and increased susceptibility to various diseases [2,3,4,5].

Regular exercise brings more benefits to the body, reducing the risk of cardiovascular, respiratory, and neurological diseases, certain types of cancer, and all-cause mortality [6,7,8]. However, exercise also increases the intake of polluted air into the lungs, which may have detrimental consequences on immune health.

Understanding how air pollution and exercise interact to affect immune health is a crucial research topic for public health. Previous studies have shown that exercise can protect against the damage caused by air pollution, but the underlying mechanisms remain unclear. Therefore, a key goal is to elucidate how exercise confers its protective effects and to determine safe thresholds for exercising in polluted environments. In this review, we have summarized the evidence from humans, mice, and cell experiments on the effects of air pollution and exercise, as well as their combined effects, on three aspects of immune health: inflammatory response, oxidative stress, and immune cells. Based on this evidence, we have proposed a possible mechanism of how exercise can counteract the negative effects of air pollution. We hope that this review will provide useful information and guidance for future research on this important topic.

## 2. The Harm of Air Pollution on the Immune Health of the Body

Air pollution poses a serious threat to respiratory health. The respiratory tract is not only the primary interface of exposure to air pollution, but also the main interface between the immune system and the environment. Pollutants can stimulate respiratory epithelial cells and specialist immune cells within the airways [9,10,11,12], leading to a multi-cellular immune response that disrupts immune homeostasis and increases the risk of various diseases [13,14,15,16].

Air pollution exposure is linked to various adverse health outcomes, such as asthma, chronic obstructive pulmonary disease (COPD), and lung cancer. It also affects asthma control, lung function development, COPD incidence and exacerbation, and cardiovascular disease risk [17,18,19,20]. Figure 1 depicts the trend of disease burden attributable to household air pollution from 2000 to 2017 [21]. Respiratory diseases emerge as the leading cause of mortality, constituting 75% of all DALYs. Communicable respiratory diseases represent the majority of the respiratory burden, followed by chronic respiratory diseases, lung cancer, and cardiovascular diseases [21]. In a 2018 joint article by British and American cardiovascular experts published in the *European Heart Journal*, particulate matter (PM) was identified as a key factor in the harmful effect of air pollution on the cardiovascular system. Ultrafine particulate matter (UFPM) can penetrate the blood through the lungs, induce vascular inflammation and immune response, and cause long-term immune damage that leads to atherosclerosis. This can result in acute myocardial infarction, heart failure, and arrhythmia [16,22].

Air pollution can also affect the nervous system. It is thought to be a chronic source of neurological inflammation and reactive oxygen species (ROS), which can damage the central nervous system through various cellular and molecular pathways and trigger innate immune responses. Neurological inflammation and ROS can lead to neuroinflammation, oxidative stress, cerebrovascular damage, and neurodegenerative pathology, with associated neurological disorders such as Alzheimer’s and Parkinson’s diseases [23]. Moreover, air pollution can cause neuronal cell damage, especially in fetuses and infants, resulting in permanent brain damage or an increased risk of neurological disease in adulthood [24]. However, the biological mechanism of air pollution in neurological diseases is still incompletely understood, and more studies are needed to elucidate the relationship between them.

Air pollution also affects the immune health of the body, both in adults and adolescents. Li et al. reported that children chronically exposed to polluted areas had decreased B lymphocyte count and C3 and C4 levels and increased monocyte count and CD8+ T lymphocyte proportion, indicating toxic effects on immunity and adaptive responses [25]. Prunicki et al. also found associations between air pollution exposure and methylation of immunoregulatory genes, immune cell profiles, and blood pressure, suggesting that the immune and cardiovascular systems were negatively impacted by air pollution even at a young age [26]. Moreover, exposure to air pollution during different critical windows of early life may alter cellular immune responses differently [27]. Azahara et al. found that prenatal exposure to traffic air pollution during pregnancy, especially in early and late gestation, impaired fetal immune system development by disturbing cord blood leukocyte and lymphocyte distributions [28,29]. In summary, excess pollutants can disrupt the body’s antioxidant defense system, leading to increased oxidative stress and inflammatory responses. This can further suppress immune cell activity and affect the multi-cellular immune response. This long-term accumulation can cause systemic, chronic low-grade inflammation, which in turn affects the immune health of the body and increases the risk of developing respiratory, cardiovascular, as well as neurological diseases, and certain cancers.

## 3. Air Pollution Induces Oxidative Stress and Inflammatory Responses and Suppresses Immune Cell Activity

Pollutants impair immune health through multiple mechanisms that are not fully understood. PM and gaseous components (CO, NO_2_, O_3_, SO_2_) constitute ambient air pollutants [30]. PM can stimulate cells through TLR receptors and trigger immune responses [30]. PAH molecules in PM can also sense AhR receptors, exacerbating immune cell responses, activating pro-inflammatory intracellular signaling cascades, and triggering cell apoptosis and immune response imbalance [31,32]. Heavy metal components in PM and gaseous pollutants also directly induce NOX, NOS, and cytochrome P-450 to produce a large amount of ROS [33,34], thereby activating the MAPK cascade and increasing intracellular Ca^2+^ concentration, aggravating inflammation, as well as damaging health [31,35]. In addition, elevated ROS levels contribute to lipid peroxidation, further exacerbating inflammation within the body [22]. The primary transcription factor enhancing antioxidant expression, Nrf2, is predominantly regulated by the protein KEAP1 [32]. The KEAP1-NRF2 system is an important cellular defense system that can detoxify chemicals and heavy metals adsorbed on environmental particulate matter and regulate the activity of antioxidant enzyme synthesis, reducing ROS levels [32] (Figure 2).

### 3.1. Exposure to Air Pollution Induces Oxidative Stress

Redox homeostasis consists of a complex network of molecules, signaling proteins, and enzymes that maintain the balance between the production and metabolism of ROS and reactive nitrogen species (RNS) in and around cells [36]. Antioxidant systems (e.g., the Nrf2-controlled gene pathway, the glutathione system, and nonenzymatic antioxidants) can regulate ROS levels [36,37]. The accumulation of ROS that exceeds the capacity of the cellular antioxidant system is often referred to as oxidative stress [36,38]. Oxidative stress is a common consequence of pollutant exposure that disrupts the balance between oxidation and antioxidation and damages cellular antioxidant defenses [39]. PM induces oxidative stress by generating ROS and free radicals in both cellular and non-cellular systems. Several sources of oxidant production have been identified, including NADPH oxidase (NOX), cytochrome P-450, and nitric oxide synthase (NOS). The composition of PM (especially heavy metals and organic compounds) is also implicated in oxidative stress induction [33,34]. For example, a study comparing winter PM_2.5_ between Beijing and Guangzhou showed that Beijing PM_2.5_ had higher levels of metals and PAHs per mass unit and caused more oxidative stress in vitro [40]. PAHs can also interact with inflammatory and antioxidant factors via the aryl hydrocarbon receptor (AhR) [41]. Moreover, PM_2.5_ exposure increases malondialdehyde (MDA), a marker of oxidative stress [42], which inhibits Nrf2 signaling and reduces the expression of antioxidant enzymes such as HO-1 and superoxide dismutase (SOD) [43]. NOX activation in the aorta also contributes to oxidative stress. Thus, PM_2.5_ impairs antioxidant capacity, enhances lipid peroxidation, and triggers oxidative stress [44,45].

PM_2.5_ exposure induces ROS that causes oxidative stress and activates the NOD-like receptor protein 3 (NLRP3) inflammasome, resulting in increased expression of caspase-1, IL-1β, and IL-18, which enhance pulmonary responses. PM_2.5_-induced ROS also correlates with activation of transient receptor potential melastatin 2 (TRPM2) and Ca^2+^ influx [46]. Oxidative stress can trigger ryanodine receptor 2 (Ryr2)-Ca^2+^ signaling, disrupt calcium homeostasis, and promote cardiac injury in hyperlipidemic mice [47]. Haberzettl et al. report that short-term PM_2.5_ exposure induces vascular insulin resistance and inflammation through a mechanism involving pulmonary oxidative stress [48].

In addition, O_3_ and NO_2_ are oxidants that often coexist with PM in ambient air and can cause oxidative damage to the body. Studies have shown that NO_2_ inhalation at different concentrations induces mild pathological changes in rat hearts, reduces or increases the activity of antioxidant enzymes (such as Cu/Zn-SOD, Mn-SOD, and glutathione peroxidase [GPX]), and enhances the formation of MDA and protein carbonyl (PCO), leading to oxidative stress [49]. O_3_ exposure may trigger ROS accumulation through lipid peroxidation of pulmonary surfactant phospholipids and cell membranes. ROS then rapidly activates the release of alarmins such as IL-1β, IL-6, IL-23, tumor necrosis factor-α (TNF-α), and thymic stromal lymphopoietin (TSLP), causing pro-inflammatory changes in the respiratory mucosal tissue structure and immune cells [50]. In a similar study of O_3_ exposure in healthy adults, long-term, 2-week O_3_ exposure was positively associated with proinflammatory cytokines; that is, sustained O_3_ exposure up-regulated redox homeostasis and enhanced the pro-inflammatory state [51]. In mouse studies, O_3_ exposure has been shown to modulate vascular tone and increase oxidative stress and mitochondrial DNA damage, with adverse effects on the respiratory and cardiovascular systems [52].

### 3.2. Exposure to Air Pollution Induces Inflammatory Responses

Ambient PM can stimulate the body to produce a large number of inflammatory cytokines, trigger local inflammation, activate relevant inflammatory effector receptors, and induce systemic inflammatory cascade reactions through the homing of immune cells. The long-term accumulation of such changes can lead to systemic low-grade inflammation, increasing the risk of cardiovascular disease [53,54], obesity [55], insulin resistance [56], Alzheimer’s disease, depression [57,58], and other disorders [59].

Previous studies have reported that atmospheric PM induces inflammation through the TLR pathway, and both biotic and abiotic lipopolysaccharide (LPS) in PM_2.5_ can activate through TLRs [60]. PM can stimulate cells through the TLR4 pathway, which in turn activates pro-inflammatory intracellular signaling pathways, such as nuclear factor-kappa B (NF-κB) and mitogen-activated protein kinase (MAPK) pathways [35,61]. For example, Fashi et al. reported that PM_10_ exposure increased TLR4 gene expression and NF-κB-dependent inflammatory responses in rat lung tissue [62]. Co-exposure to PM_2.5_ and SO_2_ also enhanced NF-κB, phosphorylated P38, and toll-like gene expression, causing inflammatory injury in rat lungs [63]. ROS generated by metals and organic matter in PM can induce oxidative stress, which can activate the MAPK cascade (ERK, p38, and Jun kinases), leading to NF-κB and activator protein-1 (AP-1) signaling and pro-inflammatory cytokine release [33].

In addition, O_3_ exposure stimulates epithelial cells to produce pro-inflammatory cytokines [64]. Both acute and chronic O_3_ exposure induces a sustained inflammatory response in the lung, characterized by neutrophilia, elevated levels of cytokine and chemokine mRNA, and protein, in bronchoalveolar lavage fluid (BALF). This is associated with the loss of antioxidant Nrf2 and SOD activity, enhanced intracellular oxidative stress, and increased HIF-1α signaling. Long-term O_3_ exposure can also induce changes in lung morphology [65]. Similarly, after exposure to different concentrations of NO_2_, the mRNA and protein expressions of inflammatory markers (TNF-α, IL-1β, ICAM-1) were up-regulated and correlated with cardiac injury in rats [49]. TNF-α and IL-1β are two typical pro-inflammatory cytokines that initiate immune responses, enhance endothelial cell permeability, and activate the expression of other pro-inflammatory genes, such as ICAM-1 [66]. NO_2_ also induced an imbalance in the ratio of Th1/Th2 differentiation (IL-4, IFN-γ, GATA-3) and the activation of the JAK-STAT pathway [67]. There are many studies on the inflammatory response induced by SO_2_ exposure [63,68,69]. For instance, Li et al. [69] explored the effects of SO_2_ exposure on inflammation and immune responses in asthmatic rats. They discovered that SO_2_ exposure overexpressed the pro-inflammatory factors, aggravating the inflammatory response in the lung, and causing Th1/Th2 imbalance. Airway inflammation and immune response increase the risk of asthma.

### 3.3. Effects of Air Pollution on Different Immune Cell Types

Immune cells have different roles in immunity. Macrophages and neutrophils are innate immune cells that respond quickly to inhaled pollutants, and dendritic cells trigger a complex adaptive immune response when they encounter pathogens [70]. The specific functions and roles of immune cells are illustrated in Table 1. These immune cells do not act alone but coordinate with each other to mount an effective defense. However, when this coordination fails or becomes excessive, it can lead to diseases [30].

#### 3.3.1. Macrophages

Alveolar macrophages (AMs), which are key players in clearing and responding to inhaled PM, can trigger immune activation. PM contains various components, such as endotoxins, transition metals, and PAHs, which can stimulate macrophages to produce ROS and pro-inflammatory cytokines (such as TNF-α and IL-1β). ROS and pro-inflammatory cytokines activate the innate immune system and contribute to pulmonary inflammation [71]. Previous studies have shown that exposure to air pollution particles selectively impairs the surface receptors and phagocytic capacity of AMs. Moreover, this impairment is more pronounced for AMs exposed to PM_10_ than PM_2.5_ [72,73].

In addition to PM, O_3_ exposure induces AMs to release cytokines and fibronectin, triggering inflammation. O_3_ also alters TLR4 distribution and enhances the endotoxin response in AMs. Moreover, O_3_ inhibits AM-mediated immunosuppression in vivo by releasing soluble factors that block NO production [74]. Similarly, NO_2_ exposure impairs the immunosuppressive function of macrophages [75,76].

#### 3.3.2. Dendritic Cells (DCs) and Lymphocytes

DCs bridge innate and adaptive immunity by recognizing antigens through innate receptors and presenting them to T lymphocytes in antigenic complexes. This activates T cells to generate antigen-specific memory and mount efficient responses to novel pathogens [77].

Multiple studies have shown that PM stimulates DCs to mature [78,79], and enhances T cell responses [78,80]. Porter et al. also found that diesel exhaust particles (DEPs) inducement of DCs results in enhanced T cell cytokine production in allogeneic co-cultures of DCs and CD4 lymphocytes [78]. In addition, in studies of naïve T cells, UPM-treated DCs have been reported to enhance the proliferation of naïve T cells [79]. PM activates DCs in an AhR-dependent manner and primes naïve T cells [81] to differentiate into inflammatory Th1, Th2, and Th17 effector cells [82] that produce pro-inflammatory cytokines [83].

O_3_ exposure also increases the level of activated DCs in the thoracic lymph nodes and promotes allergic sensitization through TLR4-dependent pathways [84]. NO_2_ inhalation induces the maturation of pulmonary CD11c+ cells that increase cytokine production and enhance their ability to stimulate naïve T cells [85].

Most of the adaptive immune effects of air pollution are focused on the T lymphocytes. B lymphocytes are another important type in the adaptive immune response, and upon stimulation, they mature into plasma cells that secrete antibodies or immunoglobulins. For example, Zhao et al. reported that Shanghai traffic police exposed to high levels of PM_2.5_ showed changes in immunoglobulin levels: a decrease in IgA levels, and increases in IgM, IgG, and IgE levels [86].

#### 3.3.3. Granulocytes

Granulocytes, such as neutrophils and eosinophils, are innate immune cells that can rapidly kill pathogens by releasing proteases and oxidases. They recognize foreign substances and pathogens via innate receptors, such as toll-TLRs. Pollutant exposure activates neutrophils and recruits them to sites of inflammation.

Several studies [87,88,89] have shown that pollutant exposure increases neutrophils and eosinophils in the peripheral blood and BLF. For example, Salvi et al. found that healthy volunteers exposed to diluted diesel exhaust had elevated neutrophils and platelets, indicating a systemic and pulmonary inflammatory response [88]. Bendtsen et al. observed a dose-dependent increase in neutrophils and eosinophils on day 1 post-exposure in mice after one intratracheal infusion of airport emission particles [89]. Other studies [90,91,92] have compared the effects of PM_2.5_ and PM_10_ on mice and found that PM_10_ exposure caused more significant changes in neutrophil numbers, possibly due to higher endotoxin and metal content.

NO_2_ exposure also induced neutrophilia in the bronchial wash samples of healthy individuals [93]. The expression of cytokines IL-5, IL-10, IL-13, and ICAM-1 increased following NO_2_ exposure, and continued to stimulate neutrophils [94]. However, the effects of pollutants on eosinophil inflammation and the direct mechanisms by which pollution affects neutrophil function are poorly understood.

## 4. Exercise-Mediated Protection against Air Pollution-Induced Immune Damage

Regular exercise boosts immune surveillance and confers various health benefits over time. It also preserves or improves immunity, possibly by modulating anti-inflammatory and antioxidant responses and enhancing immune cell function through unknown mechanisms [95] (Figure 3). There is also a large body of research showing that the beneficial effects of long-term regular exercise and fitness can counteract the immune damage caused by pollutants. We searched databases including MEDLINE, PubMed, Web of Science, and Google Scholar with keywords including air pollution, physical exercise, inflammation, oxidative stress, immunity, etc. We focused on population-based studies examining the effects of air pollution and exercise on inflammation and oxidative stress in the body (Table 2). However, research into the protective effects of exercise is still in its infancy, and it appears that exercise can improve the immune damage caused by pollutants by modulating oxidative stress and inflammatory responses.

### 4.1. Exercise Antioxidant Inhibits Pollution-Induced Oxidative Stress

Research on the relationship between exercise, oxidative stress, and antioxidant defense has been ongoing for decades. Variable exercise types, durations, and intensities have variable effects on the generation of ROS in the body [36]. Generally, vigorous acute exercise and muscle contractions lead to increased production of ROS and RNS, promoting oxidative stress in skeletal muscles [37,105]. The sources of ROS during exercise, which have been extensively documented, are primarily generated by mitochondria and related enzymes (NADPH oxidase, NOX4, NOX2, xanthine oxidase) [22,36,37]. Regular moderate-intensity exercise, on the other hand, can increase the expression and activity of endogenous antioxidant enzymes (SOD, GPX, CAT) in skeletal muscles, thereby improving the capacity to mitigate adverse reactions caused by ROS [36,105,106]. The enhanced antioxidant capacity of muscle cells also contributes to reduced ROS levels. Zarrindast et al. [107] noted that eight weeks of moderate-intensity aerobic training on land and in water reduced oxidative stress and improved antioxidant status. Done et al. [108] concluded that regular aerobic exercise increased resistance to oxidative stress. Estébanez et al. [109] demonstrated that aerobic exercise did not lead to significant changes in oxidative stress biomarkers in older adults. In animal studies, mice distributed to perform long-term swimming training showed higher levels of CAT, SOD, and GPX in their bodies [110], while rats exhibited increased levels of SOD and catalase in muscles after running [111].

Furthermore, exercise activates PGC1-α, which plays a role in improving mitochondrial biogenesis, and anti-inflammatory as well as antioxidant effects [36]. The activation of PGC1-α is mediated by upstream kinases such as AMPK and p38 MAPK [36,112]. Allopurinol, a xanthine oxidase inhibitor, reduces ROS production in rat muscles, thereby attenuating exercise-induced PGC1-α gene expression and downstream mitochondrial biogenesis-related transcriptional activity [113]. ROS generated during exercise also induces antioxidant enzymes, primarily through the Keap1-Nrf2-ARE pathway [114]. To cope with oxidative stress, cysteine residues on Keap1 are modified, leading to complex dissociation and Nrf2 translocation to the nucleus. In the nucleus, Nrf2 activates the transcription of antioxidant enzymes by binding to antioxidant response elements (ARE). Merry and Ristow [115] demonstrated in Nrf2 knockout mice that Nrf2 was necessary for increased expression of antioxidant genes such as SOD1, SOD2, and catalase, typically observed after a single bout of exercise, with increased SOD activity also observed after endurance training. Additionally, other redox-sensitive kinases such as AMPK, ERK, and JNK can also activate Nrf2 [116,117].

This adaptation to ROS may lower the risk of ROS-related diseases through exercise, including some cardiovascular diseases, stroke, Alzheimer’s disease, and certain cancers [118,119]. Aerobic exercise reduced ROS, improved coronary artery and vascular stiffness, and lowered the incidence of ROS-related disease [118]. Additionally, lowering oxidative stress may decrease cancer risk, and exercise reduces the recurrence of breast, colon, and prostate cancers. The molecular mechanisms of the anti-cancer effects of exercise are unknown, but enhanced expression of antioxidant genes may contribute [119]. Also, physical activity is associated inversely with Alzheimer’s disease and Parkinson’s disease incidence. Exercise training slows both diseases’ progression, and its benefits may involve increased cerebral blood flow and brain antioxidants [120].

Pollution exposure can induce ROS and free radicals that cause oxidative damage to the lungs. Regular exercise can reduce the level of ROS and scavenge free radicals by increasing the activity of antioxidant enzymes. Several animal studies have shown that aerobic exercise can protect the lungs from pollution-induced oxidative stress [121]. For example, high-intensity swimming increased lung CAT, GSH-t, and NPSH levels in mice exposed to DEP, reducing DEP-induced oxidative damage [122]. Similarly, endurance training or high-intensity intermittent exercise in mice exposed to PM_2.5_ suppressed the inflammatory response and oxidative stress caused by PM_2.5_ [123,124]. Moreover, aerobic interval training reduced the levels of inflammatory cytokines and increased the levels of antioxidant enzymes in the lungs of mice exposed to different concentrations of acute PM_2.5_, improving lung function and preventing lesion progression [125]. Similarly, two weeks of exercise training had a protective effect against oxidative stress induced by O_3_ exposure in rats, possibly due to changes in GPx and its peroxidase enzyme [126]. These results suggest that regular exercise can enhance the antioxidant defense system and mitigate the oxidative damage caused by pollution exposure in the lungs [127].

### 4.2. Exercise Reduces Inflammation and Inhibits Pollution-Induced Inflammatory Responses

Acute inflammation is a normal immune response to infection and trauma, but chronic, low-grade systemic inflammation underlies various diseases. Regular exercise may lower disease risk by exerting anti-inflammatory effects [128] (Figure 3). Firstly, regular physical exercise can reduce abdominal and visceral fat, leading to a decrease in circulating pro-inflammatory adipokines such as IL-6, TNF, retinol-binding protein 4, and leptin [129]. Secondly, during exercise, IL-6 produced by muscle contractions plays a crucial role. It can enhance glucose uptake and activate AMPK, promoting fat breakdown and oxidation [130]. Moreover, exercise-induced IL-6 can lead to subsequent increases in circulating levels of anti-inflammatory factors such as IL-10 and IL-1ra [131]. It also induces negative feedback inhibition of TNF-α expression [132], thereby reducing the levels of circulating inflammatory cytokines. Additionally, research suggests that IL-6 stimulates the release of cortisol from the adrenal glands. Cortisol is known to have powerful anti-inflammatory effects [133], and catecholamines can also suppress inflammatory adipokines induced by LPS in immune cells [134]. Finally, activation of the hypothalamic-pituitary-adrenal (HPA) axis and the sympathetic nervous system (SNS) during exercise can also result in increased circulating levels of cortisol and adrenaline [38].

TLRs play an important role in mediating systemic inflammation as they recognize endogenous molecules released in response to cell injury and necrosis, and activation of TLR signaling drives increased expression and secretion of pro-inflammatory cytokines [135]. It has been found that exercise reduces TLR expression in monocytes and macrophages, attenuating their downstream inflammatory cascade [136,137]. Simpson et al. showed that intense aerobic exercise decreased TLR2, TLR4, and HLA expression in human monocyte subpopulations [138]. Reduced expression of TLRs inhibits their downstream responses [139] and reduces the infiltration of macrophages into adipose tissue, thereby reducing the production of inflammatory cytokines [140]. Furthermore, exercise can increase regulatory T cells and reduce the ratio of inflammatory monocytes and classical monocytes, all of which contribute to reducing inflammation in the body [141,142].

Long-term exposure to pollution can cause chronic inflammation in the body, but regular exercise can attenuate its adverse effects. Silveira et al. examined the inflammatory markers of 10 cyclists who performed a 50 km cycling time trial under traffic air pollution and found no evidence of inflammation induced by pollution in cyclists who were adapted to it [103]. Similarly, Dos Santos et al. compared the immune/inflammatory responses in the respiratory tract of street runners and sedentary people after acute and chronic PM exposure and found that regular outdoor endurance exercise improved the respiratory immune/inflammatory status compared with a sedentary lifestyle [143]. Moreover, Rundell et al. reported that high-intensity exercise under traffic air pollution exposure did not alter the anti-inflammatory balance (IL-10/TNF-α ratio), despite a 1.7-fold increase in IL-6 concentrations, suggesting that exercise suppressed the expected anti-inflammatory response to pollution exposure [102].

Also, in animal studies, long-term aerobic training inhibited DEP-induced lung inflammatory cytokines and ROS levels in mice and had a protective effect against lung inflammation [121]. Moreover, treadmill training in mice suppressed the accumulation of immune cells and levels of pro-inflammatory cytokines in the lungs induced by particulate matter, thereby suppressing the particulate-induced systemic inflammatory response [144]. It has also been found that regular exercise for 4–8 weeks before the start of the air pollution cycle can be effective to some extent in counteracting the decline in lung function and inhibiting the development of inflammation caused by air pollution [145,146]. Pre-exercise also increased intracellular iHSP70 levels, decreased eHSP70 levels, and reduced NF-κB inflammatory pathway protein expression, suggesting that pre-exercise can counteract respiratory inflammation caused by PM in aged rats. This is because intracellular iHSP70 has an anti-inflammatory function through a mechanism that blocks NF-κB activation, whereas extracellular eHSP70 has a pro-inflammatory effect. The altered eHSP70/iHSP70 balance caused by chronic exercise may lead to one of the key mechanisms for the diminished chronic inflammatory state of the body, increased resistance to stress injury, and enhanced cell survival. Therefore, the HSP70-TLR-NF-κB signaling pathway mediated by eHSP70/iHSP70 may be one of the main mechanisms of respiratory inflammation caused by atmospheric PM [124]. In addition, HSP70 can also inhibit JNK activation [35]; therefore, it could be hypothesized that HSP70 could reduce the release of pro-inflammatory cytokines by inhibiting the activation of JNK. Similarly, oxidative stress can activate the MAPK cascade to promote the release of inflammatory cytokines [33], and it is also hypothesized that exercise could reduce the release of pro-inflammatory cytokines by reducing ROS levels and inhibiting MAPK phosphorylation. However, these mechanisms need to be further investigated in subsequent studies.

### 4.3. Effects of Exercise and Air Pollution on Immune Cells

Exercise can boost the activity and phagocytosis of macrophages, which are immune cells that engulf and digest pathogens and debris. Silveira et al. performed an acute swimming experiment on sedentary rats and found a 2.4-fold increase in the phagocytic capacity of macrophages [147]. Sugiura et al. investigated the effects of different daily exercise durations on macrophage function in mice and showed that running for at least 30 min a day was effective in enhancing macrophage function in mice [148]. In addition, chronic exercise training can inhibit macrophage infiltration into adipose tissue, increase the expression of M2 markers, reduce the expression of TNF-α and TLR4 mRNA, and reduce the inflammatory response [140].

DCs are antigen-presenting cells that activate T cells and initiate adaptive immune responses. Exercise can modulate DC function in different ways. For example, a single session of exercise in healthy adults increased the production of monocyte-derived DCs [149] and the number of circulating DCs [150]. Similarly, cyclic endurance training enhanced the maturation of dendritic cells [151].

Exercise can also affect T cell function and senescence. Moderate exercise reduced the number of senescent T cells [152] and improved T cell quality and function [153,154]. Spielmann et al. found that higher aerobic fitness was associated with more naïve and fewer senescent T cells [152]. They suggested that regular exercise may induce selective apoptosis of senescent T cells, allowing them to be replaced by new T cells that can respond to new antigens [155].

Neutrophils are the most abundant type of white blood cells and play a key role in innate immunity. Exercise can influence neutrophil numbers and function in various ways. For instance, long-term exercise training was associated with a lower neutrophil count in older men [156]. Syu et al. showed that acute, severe exercise induced neutrophil apoptosis, while chronic regular exercise enhanced neutrophil chemotaxis, phagocytosis, and citrate synthase activity and delayed their apoptosis and risk of infection [157]. Moreover, Chen et al. reported that hypoxic exercise training increased the expression of neutrophil adhesion molecules and regulatory receptors, as well as their bactericidal and apoptotic activities after acute strenuous exercise [158]. These findings suggest that moderate exercise can reduce peripheral blood neutrophils and improve their phagocytic function and lifespan.

The combined effects of air pollution and exercise on immune cells are less understood than their separate effects. Exercise has been shown to reduce the accumulation of neutrophils and lymphocytes induced by pollutants. For instance, DEP exposure increased the total cell number, neutrophils, macrophages, and lymphocytes in BALF, but aerobic exercise inhibited lung neutrophil and lymphocyte infiltration [121]. For macrophages, studies have found that there is no difference in the number of macrophages between DEP-exposed exercise mice and non-exercise mice, but the macrophages of DEP-exposed exercise mice ingested significantly more DEP particles than those of non-exercise mice. Treadmill training suppressed PM_2.5_ and PM_10_-induced accumulation of total leukocytes [159], neutrophils, macrophages, and lymphocytes in bronchoalveolar fluid, as well as levels of other inflammatory cytokines, thereby reducing particulate-induced pulmonary and systemic inflammation [144].

Zhang et al. investigated the association between white blood cell count, physical activity, and PM_2.5_ in 359,067 Taiwanese residents. They found that long-term PM_2.5_ exposure was associated with a lower peripheral blood white blood cell count in subjects who did not regularly participate in physical activity [160]. Moreover, some researchers conducted 12 weeks of aerobic training on subjects in urban and rural areas with different UPM concentrations, which were significantly higher in urban than rural environments. They found that the urban group had an increased white blood cell count, neutrophil count, and exhaled NO, while these parameters did not change in the rural group. This suggests that training under heavily polluted conditions enhances inflammatory biomarkers [161]. Therefore, more evidence is needed to determine whether exercise can reduce leukocyte and neutrophil accumulation induced by air pollution.

## 5. Challenges and Future Directions

Chronic exposure to air pollution increases the risk of developing many diseases, and regular exercise is a powerful medicine against a wide range of diseases. Air pollution and exercise are linked through multiple physiological and behavioral mechanisms, and these have important implications for public health. The preceding sections have summarized the potential mechanisms through which air pollution and exercise impact immune health, providing evidence of their combined effects on immune health. However, research on the interaction between the two is still in its nascent stages, and there is currently no definitive answer on how to integrate exercise regimens to maximize the beneficial effects of exercise while mitigating the adverse effects of air pollution on health. Drawing from past experiences and existing research, we summarize the challenges faced by researchers and outline future directions for development.

### 5.1. Large-Scale Population Epidemiological Studies at Different Levels

Currently, global air pollution remains unabated, while more people prefer participating in outdoor activities. However, the large-scale studies on the combined exposure of exercise and air pollution on immune health remain sparse. It is imperative to conduct extensive population-based research on different demographic groups engaging in physical activity under air pollution conditions based on different people. Additionally, integrating environmental quality monitoring with medical and health data can help establish epidemiological health risk models. These models can determine safe thresholds for exercising in air pollution for various population characteristics and facilitate the development of a comprehensive exercise health management platform. This platform would offer environmental safety alerts tailored to different demographics, environments, and physical conditions.

### 5.2. Animal and Cell Experiments Exposed to Air Pollution

Due to ethical considerations, current research on long-term physical exercise under conditions of air pollution exposure relies heavily on animal and cell experiments. There is a relative scarcity of studies investigating the mechanisms by which exercise mitigates the physiological damage caused by exposure to air pollution. Further research is needed to strengthen our understanding in this area. Although animal and cellular experiments are important in understanding the potential pathological mechanisms of air pollution, these experiments may limit the information obtained from the experiments to exposure to the pollution. In cell culture experiments, the PM deposited per cm^2^ of surface may vary considerably depending on the diameter of the culture dish. It is questionable whether they represent the density of PM deposited in the airways. Many animal experiments applied traditional tracheal drip methods for contaminant exposure, which are traumatic given the large concentrations of contaminants. It is therefore recommended that studies should be conducted using whole-body exposure concentrated enrichment, which is non-invasive and better simulates realistic air pollution exposure [162,163].

### 5.3. Studies on Exposure to Mixed Pollutants Versus Single Pollutant Exposure

Air pollution is a complex mixture of various toxic substances. Currently, the literature predominantly consists of studies on exposure to mixed pollutants. Research on exposure to pollutant mixtures more comprehensively reflects the effects of pollutants on the human body and aligns better with the exposure context of air pollution [164]. However, the study of the mechanisms of mixed exposure presents significant challenges. It is suggested to increase research on exposure to single pollutants and to identify harmful components. This approach can help eliminate the influence of other pollutants and obtain more targeted experimental information, accelerating in-depth mechanistic studies. Subsequently, studies on exposure to mixed pollutants can be conducted to explore the interactive effects of multiple pollutants on human health.

### 5.4. Research on the Combined Exposure of Air Pollution and Exercise on Human Populations

Many studies on human populations compare the benefits of exercising under different air pollution concentrations on immune health (Table 2). Low air pollution exposure is often used as a baseline reference value for the comparison and analysis. However, low air pollution exposure does not equate to completely clean air, which may introduce confounding effects and impact the mechanistic evidence on the immune health effects between the two factors. Additionally, there is limited research on populations exposed to long-term air pollution and exercise, with many studies overlooking the delayed effects of interaction between air pollution and exercise. Most studies collect immediate information on the joint exposure of both factors on immune health, whereas the response of immune health to the combined exposure may require a longer time to manifest. For future exploration of the interactive effects of pollutant exposure and exercise on immune health, long-term studies on air pollution and exercise exposure should be conducted, collecting longer-term response information on immune health. Furthermore, consideration should be given to the impacts on other systems, exploration of more comprehensive health outcomes, and the design of various exposure scenarios, such as different levels of exercise intensity and modes, as well as different pollution concentration scenarios, to understand the interaction between air pollution and exercise.

## 6. Conclusions

In short, pollutants can directly affect our immune system by triggering certain signals within our cells. This can lead to inflammation and increase the risk of diseases. However, exercising moderately might help counteract some of these effects. The relationship between exercise and pollution is complex, and more research is needed to understand how much exercise is needed to protect against pollution. We also need to understand better how exercise helps the body fight pollution-related damage to the immune system.

## Figures and Tables

**Figure 1 biology-13-00247-f001:**
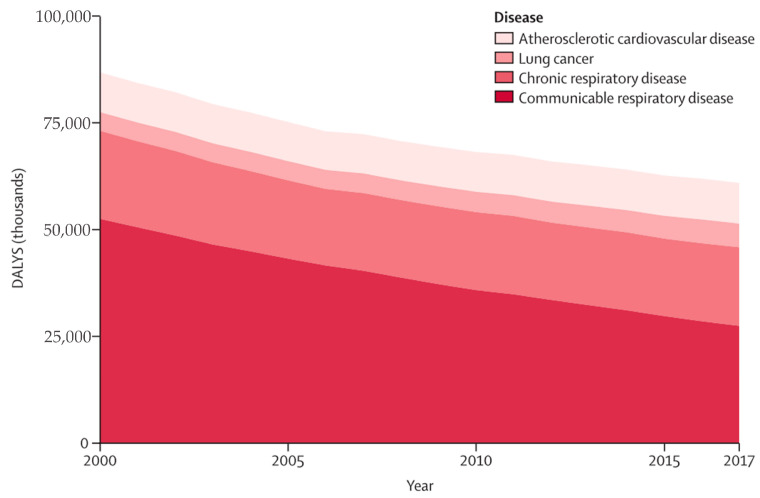
Trends in cause-specific burden of disease attributable to household air pollution, 2000–2017. DALYs = disability-adjusted life-years. Adapted from Lee et al. [21].

**Figure 2 biology-13-00247-f002:**
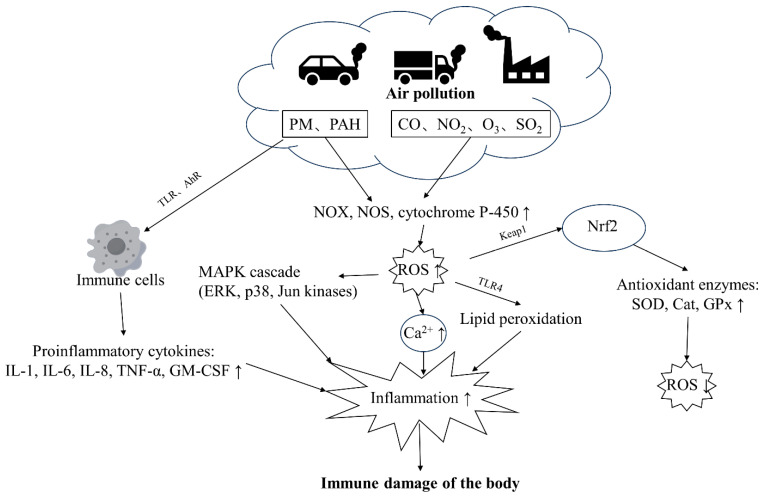
The major pathways through which air pollution damages the immune health of the body. PM and PAHs can activate immune cells through TLR and AhR receptors, leading to increased production of inflammatory factors. PM and gaseous pollutants can also stimulate oxidants, leading to increased ROS levels in the body, resulting in a cascade of reactions and increased inflammation. Abbreviation: TLR: toll-like receptor, PAH: polycyclic aromatic hydrocarbon, ROS: reactive oxygen species, NF-κB: nuclear factor kappa B, AP-1: activator protein 1, OX: NADPH oxidase, NOS: nitric oxide synthase, MAPK: mitogen-activated protein kinase, AhR: aryl hydrocarbon receptor, HO-1: heme oxygenase 1, SOD: superoxide dismutase, NLRP3: NOD-like receptor protein 3, TRM2: tropomyosin-related kinase 2, Ryr2: ryanodine receptor 2, GPx: glutathione peroxidase, TNF-α: tumor necrosis factor-α, TSLP: thymic stromal lymphopoietin, ERK: extracellular signal-regulated kinase, p38: p38 mitogen-activated protein kinase, Jun kinases: Jun N-terminal kinase, IL-1β: Interleukin-1 beta, IL-6: Interleukin-6, IL-23: Interleukin-23, IL-4: Interleukin-4, IL-8: Interleukin-8, HIF-1α: hypoxia-inducible factor 1-alpha, ICAM-1: intercellular adhesion molecule 1, IFN-γ: interferon gamma, GATA-3: GATA binding protein 3, CCL2: chemokine (C-C motif) ligand 2, HLA: human leukocyte antigen, CAT: catalase, GSH-t: glutathione transferase, NPSH: nonprotein sulfhydryl group, PGC-1α: peroxisome proliferator-activated receptor gamma coactivator 1-alpha.

**Figure 3 biology-13-00247-f003:**
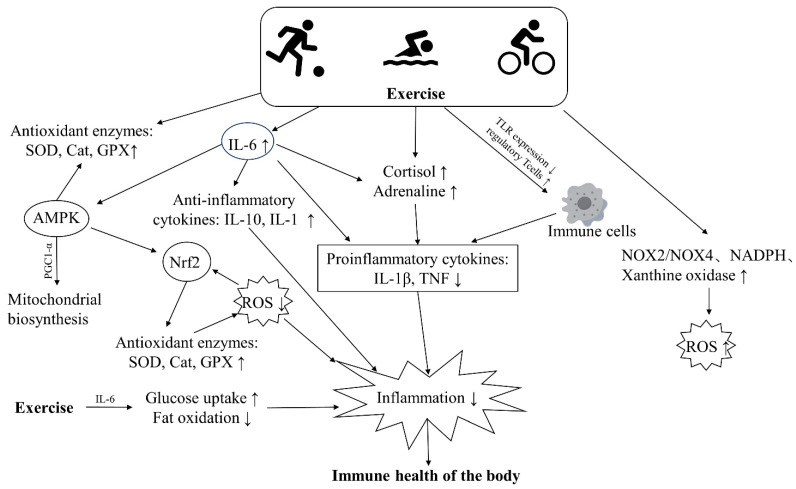
The major pathways through which exercise improves the immune health of the body. Exercise can boost the activity of antioxidant enzymes, promote fat oxidation and glucose uptake, thus lowering levels of ROS and inflammation in the body. Additionally, exercise leads to increased secretion of catecholamines, cortisol, and anti-inflammatory factors, which inhibit the release of inflammatory cytokines and alleviate inflammation. Similarly, exercise can reduce inflammation by suppressing the expression of toll-like receptors (TLRs) on cells and decreasing the release of inflammatory cytokines.

**Table 1 biology-13-00247-t001:** The functions of immune cells and their immunoregulatory roles.

	Function	Immune Regulation
Macrophages	Engulfment and digestion of foreign pathogens	Promote inflammatory responses, clear cellular debris, and activate other immune cells.
Neutrophils	Phagocytosis and killing of bacteria, viruses, etc.	Participate in acute inflammatory responses, release cytokines and chemokines, and form abscesses.
Dendritic cells	Capture and presentation of antigens	Activate T cells and B cells, initiate adaptive immune responses, regulate immune tolerance, etc.
T lymphocytes	Coordination and immune response	Differentiate into various types of T cells, participate in cell-mediated immune responses, etc.
T lymphocytes	Production of antibodies	Secrete antibodies, participate in humoral immune responses, and generate memory B cells.

**Table 2 biology-13-00247-t002:** Study on the combined exposure of air pollution and exercise on inflammation and oxidative stress in human population.

First Author/Year	Study Population	Physical Activity	Air Pollutants	Major Outcomes
Gomes, 2011 [96]	10 male athletes	8 km time trial run	O_3_	A hot, humid, and ozone-polluted environment (0.1 ppm) elicits an early epithelial damage and antioxidant protection process in the upper respiratory airways of athletes immediately after performing an 8 km time trial run.
Au, 2015 [97]	25 healthy men	exercise	acute diesel exhaust	Diesel exhaust exposure induces platelet activation. This platelet priming effect could be a contributor to the triggering of atherothrombotic events related to air pollution exposure.
Kubesch, 2015 [98]	28 healthy participants	intermittent moderate PA, consisting of four 15 min rest and cycling intervals	traffic-related air pollution (TRAP)	Intermittent moderate PA has beneficial effects on pulmonary function. Particulate air pollution can induce pulmonary and systemic inflammatory responses.
Emilia Pasalic, 2016 [99]	126 students	sports practice (0, 25th Percentile, Median, 75th Percentile)	O_3_	The moderating effects of activity level suggest that peaks of high concentration doses of air pollution may overwhelm the endogenous redox balance of cells, resulting in increased airway inflammation.
Zhang, 2018 [100]	359067 adults	Habitual PA (inactive, low, moderate, high)	PM_2.5_	Inverse association between PA and WBC; positive association between PM_2.5_ and WBC; no interaction between PA and PM_2.5._
Leonardo A Pasqua, 2020 [101]	10 healthy men	prolonged moderate exercise (i.e., 90 min)	air pollution from an urban center	The exercise of longer duration (i.e., 90 min), but not of shorter duration (i.e., <60 min), performed in vehicular air pollution conditions results in pronounced pro-inflammatory and increased arterial pressure responses.
Ramon Cruz, 2022 [102]	15 participants	High-intensity interval exercise (HIIE)	TRAP	TRAP potentially attenuates health benefits often related to HIIE. For instance, the anti-inflammatory balance was impaired, accompanied by accumulation of metabolites related to energy supply and reduction to exercise-induced decrease in SBP.
André C Silveira, 2022 [103]	10 male cyclists	50 km cycling time trial (50 km cycling TT)	TRAP	The potential negative impacts of exposure to pollution on inflammatory, neuroplasticity, and performance-related parameters do not occur in recreationally trained cyclists who are adapted to TRAP.
Li, 2023 [104]	72,172 participants	Habitual PA (inactive, low, moderate, high)	ambient particulate matter pollutants (PM_1_, PM_2.5_, and PM_10_)	Positive association with PM and risk of systemic inflammation-induced multimorbidity; positive association between PM and risk of systemic inflammation-induced multimorbidity

## Data Availability

No data were used for the research described in the article.

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
