# Peer review of "Exercise-Mediated Protection against Air Pollution-Induced Immune Damage: Mechanisms, Challenges, and Future Directions"

_biology, 2024, doi:10.3390/biology13040247_

Round 1

Reviewer 1 Report

Comments and Suggestions for Authors

This is an interesting review corellating how excercise exerts a positive effect to protect the  human organism that can be damaged from long term exposure to air polution. The whole review has an ambitious goal, though the authoirs  can reviwe some points that have to be taken into consideration in order to achieve their goal.

Mainly, there are many references missing. Lines: 53, 58, 105, 172, 222, 244, 245 (several studies????), line 253 (the text refers to "other studies " but there is only one citation), 301, 373, 457,498, 500-501.

In addition,  a small reference to the method used for the literature search and inclusion/exclusion of relative articles can enpower this review.

Author Response

Comments:
1. Mainly, there are many references missing. Lines: 53, 58, 105, 172, 222, 244, 245 (several studies????), line 253 (the text refers to "other studies " but there is only one citation), 301, 373, 457,498, 500-501. Response: Thanks for the comments. We have carefully revised the sentences and added some relevant references based on the line numbers that you have mentioned as well as other content in the manuscript. Specific revisions could be found in lines 66, 67, 198, 199, 256, 279, 284, 353, 379, and 537.

2. In addition, a small reference to the method used for the literature search and inclusion/exclusion of relative articles can this review.

Response: Thanks for the comment. We have provided a brief description of the databases used for the literature search and the keywords employed. Additionally, we summarized the studies on the effects of air pollution and exercise on the immune health of individuals. Specific revisions can be found in lines 299-303: “We searched databases including MEDLINE, PubMed, Web of Science, and Google Scholar with keywords including air pollution, physical exercise, inflammation, oxidative stress, immunity, etc. We focused on population-based studies examining the effects of air pollution and exercise on inflammation and oxidative stress in the body (Table 2).”

Reviewer 2 Report

Comments and Suggestions for Authors

Review Report of Manuscript ID Number “Biology-2918127”

The study addresses an important and timely topic concerning the potential protective effects of exercise against air pollution-induced immune damage. By incorporating evidence from human, animal, and cell experiments, the review offers a comprehensive analysis of the current understanding of this complex relationship.

The review could be improved by following suggestions;

The title could be improved or redesigned to attract more readers such as

"Exercise-Mediated Protection Against Air Pollution-Induced Immune Damage: Mechanisms, Challenges, and Future Directions" or

"Mitigating the Impact of Air Pollution on Immune Health: Exploring the Role of Exercise and Challenges Ahead"

The heading 2. The harm of air pollution on the immune health of the body could modified to respiratory system.

The author could draw a graph with time report to make a proportion between rising air pollution and respiratory diseases.  Country wise/continent wise Comorbidities of air pollution in recent times could be a better option to make it more brief and depictable and easily understandable.

The heading 3. Air pollution induces oxidative stress, and inflammatory responses and suppresses immune cell activity:

It has limitations as only two mechanisms of immune damage has been described in a brief way of one or two lines. There are some other mechanisms of immune damage and authors should describe those and include in figure 1.

The author did not describe the type of air pollution causing such damage to immune health, different concentrations of gases in air cause different toxic and immune hazards so it could be further elaborated. Figure 1

In figure 1 last box should be immune activation of body or immune hazard /damage of body as inflammation causes activation of immune system to suppress its effects.

3.1 Exposure to air pollution induces oxidative stress: the author should explain oxidative and antioxidative agents and their roles in body to make an impact of their imbalance.

169, 170, 171  line needs references The long-term accumulation of such changes can lead to systemic low-grade inflammation,  which can lead to an increased risk of cardiovascular disease, obesity, insulin resistance, 171 Alzheimer’s disease, depression, and other disorders.

The author exposed the O3 harmful effects on immune systems but did not give the other toxic gases effects like CO, NO, etc.

The paragraph of heading 3.3. Effects of air pollution on different immune cell types requires references and also elaboration about cell types of body and immunomodulatory role in body by a simple table to make it more understandable.

The section of review 4. Exercise antioxidants, reduce the inflammatory response, and enhance immune cell activity requires extensive editing and modification. Actually the whole theme of review is based on this part and this part needs to be rearranged and effectively divided into tables or graphs to show a continuous effect or known studies to ameliorate the toxic effects of pollution through exercise.

Figure 2 requires further elaboration and could be redesigned to make it clear.

4.1. Exercise Antioxidant requires latest references as these are too old. Only one reference of 2018, while all others are too old. There are very recent reports on this section need to be quoted.

4.2. Anti-inflammatory effects of exercise need to be revised. The three subheadings could be interrelated and their gradual effect could be manifested.

6. Discussion and conclusion are very poorly written and did not represent the mist of studies described in the review. It should be rearranged and rephrased.

Although the study acknowledges that the underlying mechanisms involved in the protective effects of exercise on pollutant-induced damage are still not fully understood. This limitation may affect the depth of the analysis and the ability to draw definitive conclusions about the relationship between exercise and air pollution.

The overall language of the manuscript requires editing, grammatical corrections and rephrasing. Some sections of the review has extracted from others as it is and therefore 27% similarity index while AI is too low which is a healthy sign about the originality of the manuscript. 

The review may not cover all relevant studies or may have a limited scope, potentially overlooking important research findings or alternative perspectives on the topic.

Addressing these limitations through rigorous study design, addition of tables or systematic or graphical models, and careful interpretation of results can enhance the validity and impact of the research findings.

Comments on the Quality of English Language

The overall language of the manuscript requires editing, grammatical corrections and rephrasing. Some sections of the review has extracted from others as it is and therefore 27% similarity index while AI is too low which is a healthy sign about the originality of the manuscript. 

Reviewer 3 Report

Comments and Suggestions for Authors

In the present review entitled: "Effects of exercise combined with air pollution on immunity 2 system both in human and animal models" authors discussed in a critical point of view the air pollution, a serious risk factor for human health, can lead to immune damage and  various diseases.  They highlighted the importance of exercise that has been shown to modulate anti-inflammatory and antioxidant statuses, enhance immune cell activity, as well as protect against immune damage caused by air pollution. They concluded by suggesting strengthening epidemiological studies at different population levels and investigations on immune cells to provide  guidance on how to determine the safety thresholds for exercise in polluted environments.

This is a very interesting review and I suggest to ameliorate the figures, because They seem too simple and with a poor resolution.

Round 2

Reviewer 2 Report

Comments and Suggestions for Authors

The authors have greatly improved the manuscript and majorly revised the whole manuscript. He addressed all of my questions and suggestions and made a good impact on the overall designing and composition of manuscript. I appreciate all the newly added references as well and strongly recommend the acceptance after the briefness of conclusion. the conclusion seems to me very long and complicated so i am giving an advice on it to rewrite or modify it as below:

Conclusion:

"In short, pollutants can directly affect our immune system by triggering certain signals within our cells. This can lead to inflammation and increase the risk of diseases. However, exercising moderately might help counteract some of these effects. The relationship between exercise and pollution is complex, and more research is needed to understand how much exercise is needed to protect against pollution. We also need to understand better how exercise helps the body fight pollution-related damage to the immune system".

Comments on the Quality of English Language

Minor grammatically errors still can find in the manuscript so the author need to make it error free for publication.

Author Response

1. The authors have greatly improved the manuscript and majorly revised the whole manuscript. He addressed all of my questions and suggestions and made a good impact on the overall designing and composition of manuscript. I appreciate all the newly added references as well and strongly recommend the acceptance after the briefness of conclusion. The conclusion seems to me very long and complicated.

Response: I greatly appreciate your suggestion in drafting the conclusion. The new conclusion was rewritten as “In short, pollutants can directly affect our immune system by triggering certain signals within our cells. This can lead to inflammation and increase the risk of diseases. However, exercising moderately might help counteract some of these effects. The relationship between exercise and pollution is complex, and more research is needed to understand how much exercise is needed to protect against pollution. We also need to understand better how exercise helps the body fight pollution-related damage to the immune system.”

2. Minor grammatically errors still can find in the manuscript so the author need to make it error free for publication.

Response: Thanks for the comments. Specific revisions could be found in lines 27-29, 179-183, 212-215, 242-244, 271-272, 276-277, 307-313. The revised content is as follows:

“Furthermore, we suggest strengthening epidemiological studies at different population levels and investigations on immune cells to guide how to determine the safety thresholds for exercise in polluted environments.” 

“Studies have shown that NO2 inhalation at different concentrations induces mild pathological changes in rat hearts, reduces or increases the activity of antioxidant enzymes (such as Cu/Zn-SOD, Mn-SOD, and glutathione peroxidase [GPX]), and enhances the formation of MDA and protein carbonyl (PCO), leading to oxidative stress.” 

“In addition, O3 exposure stimulates epithelial cells to produce pro-inflammatory cytokines[64]. Both acute and chronic O3 exposure induces a sustained inflammatory response in the lung, characterized by neutrophilia, elevated levels of cytokine and chemokine mRNA, and protein, in bronchoalveolar lavage fluid (BALF).”

“ROS and pro-inflammatory cytokines activate the innate immune system and contribute to pulmonary inflammation.” 

“For example, Zhao et al. reported that Shanghai traffic police exposed to high levels of PM2.5 had altered immunoglobulin levels: IgA levels, whereas IgM, IgG, and IgE levels.”

“They recognize foreign substances and pathogens via innate receptors, such as toll-TLRs.” 

“Figure 3. The major pathways through which exercise improves the immune health of the body. Exercise can boost the activity of antioxidant enzymes, promote fat oxidation and glucose uptake, thus lowering levels of ROS and inflammation in the body. Additionally, exercise leads to in-creased secretion of catecholamines, cortisol, and anti-inflammatory factors, which inhibit the release of inflammatory cytokines and alleviate inflammation. Similarly, exercise can reduce inflammation by suppressing the expression of toll-like receptors (TLRs) on cells and decreasing the release of inflammatory cytokines.”